# Studies on Modification of Polyamide 6 Plastics for Hydrogen Storage

**DOI:** 10.3390/polym17040523

**Published:** 2025-02-18

**Authors:** Li Li, Jiawei Zhao, Xingguo Wang, Qingquan Yang, Xiang Wang, Hua Yin

**Affiliations:** SINOPEC (Beijing) Research Institute of Chemical Industry Co., Ltd., Beijing 100013, China; lil.bjhy@sinopec.com (L.L.); zhaojiawei.bjhy@sinopec.com (J.Z.); wangxg.bjhy@sinopec.com (X.W.); yangqq.bjhy@sinopec.com (Q.Y.); wangxiang.bjhy@sinopec.com (X.W.)

**Keywords:** type IV hydrogen storage cylinder, permeability, polyamide modification, graphene

## Abstract

Polyamide 6 (PA6) can be used as the liner material of a type IV hydrogen storage tank, but its high hydrogen permeability and poor low-temperature toughness limit its application in related fields. In this work, PA6 composites were prepared by introducing graphene with different contents and a nano two-dimensional lamellar filler via the melt blending method, and the effects of the graphene content on the crystallization properties, mechanical properties, and hydrogen barrier properties of the composites were investigated. The results showed that composites containing 2.0 wt% graphene exhibited the best overall performance, and compared with PA6, the crystallization properties, normal-temperature impact strength and low-temperature impact strength of the composites were increased by 16.0%, 130.6% and 111.7%, respectively, and the helium permeability coefficient was decreased by 33.2%. Graphene enhances the crystallinity and lamellar thickness of the composites. Additionally, its two-dimensional lamellar structure can also increase the diffusion path of gas molecules, thereby improving the barrier properties of the composites.

## 1. Introduction

Hydrogen-fueled vehicles have a broad market, and on-board hydrogen storage technology is a key factor affecting it [1,2]. High-pressure gaseous hydrogen storage is currently the most widely used and mature hydrogen storage method [3,4,5]. Compared with the type III hydrogen storage tank, the type IV hydrogen storage cylinders utilize polymer liners to increase hydrogen storage density [6]. At present, the polymer lining materials mainly include high-density polyethylene (HDPE) and polyamide (PA). Dong et al. [7] compared the difference in the hydrogen permeability of PA6, PA11 and HDPE; the results indicate that PA6 has the best hydrogen permeation resistance because of its strong molecular polarity and hydrogen bond interaction [8,9,10,11]. Therefore, polyamide has gradually become the choice for use in a type IV hydrogen storage tank. However, the hydrogen storage material is generally expected to expose a wide range of temperature (−40 °C~85 °C) and pressure (up to 100 MPa); thus, solving the problems of high hydrogen permeability and poor toughness at low temperatures is needed [12]. The major influencing factors of hydrogen permeability include temperature, pressure and material properties [13,14,15,16]. Su et al. [17] comprehensively investigated the dissolution–diffusion behavior of H_2_ in PA6 under different temperature and pressure conditions (233~358 K, 0~87 MPa), and found that the diffusion coefficient, the solubility coefficient, and the permeability coefficient were positively correlated with temperature. Barth et al. [18] summarized the temperature dependence of hydrogen permeability of various polymer materials, and found that the hydrogen permeability coefficient of all materials increases as the temperature increases. Fujiwara et al. [19] designed a hydrogen permeability measurement system with a capacity of 90 MPa to assess the hydrogen permeability of high-density polyethylene (HDPE) at 30 °C. Their experiments revealed that the quantity of hydrogen permeating through the material increases with rising pressure, and the permeability coefficient, diffusion coefficient, and solubility coefficient of hydrogen in HDPE all decrease to varying degrees as the pressure rises. In addition to the external environment parameter, filler modification is one of the important ways to improve the barrier properties of polyamide.

The development and application of two-dimensional nanofillers have garnered significant attention in the field of material science; their high aspect ratio and large surface area have shown great potential in enhancing the barrier performance of polymers [20]. The introduction of two-dimensional nano-fillers can prolong the diffusion range of gas molecules in the matrix. Sun et al. [21] investigated the applicability of lamellar inorganic component (LIC)-modified PA6 composites as liner materials for type IV hydrogen storage. The results show that PA6/LIC exhibits superior tensile strength, bending strength, and bending modulus compared to PA6, increased by 36%, 17% and 12%. Especially of note, the hydrogen permeability of PA6/LIC is one-third that of PA6. Li et al. [22] used a molecular dynamic simulation to study the adsorption and diffusion process of hydrogen in a graphene (Gr.)-modified PA6 system, and investigated the effects of temperature, pressure on hydrogen diffusion. The results show that at atmospheric pressure, the gas permeability coefficient of PA6/5.0 wt% Gr. is 2.44 × 10^−13^ cm^3^·cm/(cm^2^·s·Pa), which is decreased by 54.6%. Takeo et al. [23] described a clay-based film designed for hydrogen tanks, which demonstrated exceptional hydrogen gas barrier properties and high durability. The hydrogen gas barrier performance of this clay-based film was about two-to-three orders of magnitude superior to that of pure polymer materials. In addition, films containing 70% clay exhibited remarkable performance in both gas barrier properties and elongation at break. In short, the current research mainly focuses on the barrier properties of two-dimensional lamellar materials on hydrogen storage material, but little attention is paid to the toughness effects of composite materials at low temperature. Hence, to balance the low hydrogen permeability and the toughness of the modified material is still an ongoing challenge in liner material research.

In this work, graphene is introduced into the PA6 matrix to prepare PA6/Gr. nanocomposites, and the effects of graphene content and dispersion on the crystalline properties, mechanical properties, gas barrier properties, toughness at low temperature of composites were investigated. This work is a reference for the development of type IV hydrogen storage liner materials.

## 2. Experiments

### 2.1. Preparation of PA6/Gr. Nanocomposites

PA6, as the polymer matrix, with a trademark of BL1340, was obtained from Sinopec Baling Petrochemical Co., Ltd. (Yueyang, China); Graphene (Gr.) as the reinforcement was obtained from Sinopec Catalysts Branch (Beijing, China); Antioxidant 1010 and Antioxidant 168 were purchased from BASF SE (Ludwigshafen, Germany). The composition of PA6/Gr. nanocomposites blends was shown in Table 1.

PA6 was dried in an oven at 80° C for 8 h. Graphene with different mass fractions was weighed. The PA6/Gr. composites with different graphene contents were prepared by melt blending through a twin-screw extruder. The temperature of each zone of the twin-screw extruder was set at 230 °C, 240 °C, 245 °C, 245 °C, 240 °C, and 240 °C; the rotational speed was set at 300 r/min. After the twin-screw extrusion, the extrudate was hauled, cooled by water, and sent to the pelletizer for pelletizing. The pellets were dried in the oven at 80 °C for 8 h. Finally, the pellets were injection-molded into standard mechanical test samples and flakes for helium permeation test with the injection temperature set at 245 °C, 245 °C, 235 °C, 235 °C, and 230 °C. The injection pressure was 58 MPa and the holding pressure was 58 MPa. 

### 2.2. Testing and Characterization

#### 2.2.1. Thermogravimetric Analysis (TGA)

TGA (TGA8000, Mettler Toledo, Greifensee, Switzerland) was used to measure the thermal weight loss of materials. The samples were weighed at about 5–10 mg, and the setup procedure was given as follows: holding at 50 °C for 1 min in a nitrogen atmosphere at a temperature-rising rate of 10 °C/min from 50 °C to 800 °C, with a nitrogen flow rate of 20 mL/min.

#### 2.2.2. Differential Scanning Calorimetry (DSC)

DSC (DSC 8000, PerkinElmer Inc., Springfield, IL, USA) was used to determine the crystallinity and melting point of the material, with a sample of about 5–10 mg; the setup procedure was as follows: held at 50 °C for 1 min, the temperature raised from 50 °C to 260 °C under nitrogen atmosphere at a rate of 10 °C/min, and held at 260 °C for 3 min to eliminate the thermal history. The temperature then dropped from 260 °C to 50 °C at 10 °C/min, followed by a second heating process with the nitrogen gas flowing at a rate of 20 mL/min. For the test of glass transition temperature (Tg), the setup procedure was as follows: after eliminating the thermal history, it was cooled to 30 °C, and then the sample was heated to 120 °C at 20 °C/min.

#### 2.2.3. Scanning Electron Microscopy (SEM)

SEM (S-4800, Hitachi, Tokyo, Japan) was used to study the microscopic morphology of the material and the dispersion of graphene in PA6. The low-temperature notched impact section samples were sprayed with gold in a gold-spraying machine for 1 min. The section morphology was observed at different magnifications.

#### 2.2.4. Transmission Electron Microscopy (TEM)

TEM (HT 7800, Hitachi, Tokyo, Japan) was used to study the dispersion of the filler; the samples were sliced in a frozen ultrathin slicer with the temperature set in advance to a thickness of about 70 nm. The slices were cut out on a knife edge and then transported with tweezers on a copper mesh moistened with alcohol or by using a sampling ring moistened with a saturated concentration of sucrose, and then affixed to the copper mesh. The filming voltage was 80 kv and the morphology was observed at different magnifications.

#### 2.2.5. X-Ray Diffraction (XRD)

XRD (D8 Discover, Bruker, Karlsruhe, Germany) was used for the structural analysis of the material and the crystallinity of the material. The tube voltage and current were maintained at 40 kV and 40 mA, respectively, with CuKα-ray (wavelength 0.1542 nm). The scanning speed was 10 °/min and data were recorded from 5 ° to 90 ° (2θ).

#### 2.2.6. Small-Angle X-Ray Scattering (SAXS)

SAXS (Nanostar, Bruker, Karlsruhe, Germany) was used for lamellar crystal analysis of polymers, with a tube voltage of 45 kV, a tube current of 0.65 mA, CuKα-ray (wavelength 0.1542 nm), and a two-dimensional faceted detector with a resolution of 2048 × 2048, a pixel size of 68 μm × 68 μm, and a specimen-to-detector distance of 1046.5 mm.

#### 2.2.7. Mechanical Properties

For tensile performance, injection molding of dumbbell-type specimen samples (length 150 mm, width 15 mm, thickness 4 mm) was carried out with a universal testing machine, according to the method specified in GB/T 1040.1-2010 [24]: a tensile rate of 10 mm/min, test room temperature of 25 °C, and relative humidity of 50%. The tensile strength and nominal strain were recorded. For flexural strength, the injection-molded specimen (100 mm ∗ 10 mm ∗ 4 mm) was tested in accordance with GB/T 9341-2008 [25]. The bending strength and bending modulus were recorded. For impact strength, the sample (100 mm ∗ 10 mm ∗ 4 mm) was notched with a 2 mm V-notch and tested at room temperature and −40 °C with relative humidity at 50%. The impact energy data were recorded.

#### 2.2.8. Helium Gas Permeability Test

The VAC-V2 Differential Pressure Method Gas Permeation Tester (Jinan Labthink International, Inc., Jinan, China) was used to test helium permeation under the conditions of 23 °C and 0% RH according to GB/T1038.1-2022 [26]. The gas permeates from the high-pressure side to the low-pressure side under the action of the differential pressure gradient, and the monitoring of the pressure within the low-pressure side is processed in order to derive the various barrier parameters of the samples.

## 3. Results and Discussion

### 3.1. Thermal Properties of PA6/Gr. Nanocomposites

The TGA curves of the corresponding PA6/Gr. nanocomposites are shown in Figure 1. The temperature of the heat loss curves ranges from 50 °C to 800 °C. And the temperature at 5% weight loss is T_5_, the temperature of maximum weight loss rate is T_MAX_, and the remaining mass fraction at 800 °C is the residual carbon rate. The data are shown in Table 2.

The thermal weight loss curves show that PA6 has the lowest residual mass of 0.66%, indicating that at 800 °C, PA6 is mostly degree-dated and vaporized at 800 °C. In PA6/Gr. composites, the residual mass is close to the added graphene content, indicating that a more stable content of graphene is dispersed into the PA6 matrix. Compared with those of PA6, the temperature at 5.0 wt% weight loss (T_5_) and the temperature of maximum weight loss rate (T_MAX_) of PA6/Gr. increased as the content of graphene increased, indicating that the composites are more thermally stable after the introduction of graphene.

The DSC curves of PA6/Gr. nanocomposites are shown in Figure 2. Figure 2a presents the melting process of the samples and Figure 2b presents the crystallization process of the samples. From the curves, the melting peak temperature, enthalpy of melting, crystallization peak temperature, and enthalpy of crystallization of the material can be obtained. Also, the DSCs are used to test the glass transition temperature (Tg) with the procedure given above. Through Equation (1), ∆*H_m_* is the enthalpy of melting of the material, and ∆*H** the enthalpy of melting of PA6 at a 100% theoretical degree of crystallinity 230 J/g [27]. The crystallinity of the material can be calculated as shown in Table 3.(1)$$X_{C}=\frac{\Delta H_{m}}{\Delta H*}\times 100\%$$

The melting curves showed that the T_m_ of PA6 was 219.64 °C, while the T_m_ of PA6/Gr. composites was slightly higher than that of PA6, with the highest T_m_ of 220.80 °C for a graphene content of 2.0 wt%. The crystallization curves show that the T_c_ of PA6/Gr. composites is significantly higher than that of PA6 with the same thermal history, indicating that the PA6/Gr. composites have a greater tendency to crystallize. The crystallinity of the PA6/Gr. nanocomposites increased as the content of graphene increased, which is due to the fact that graphene acts as a nucleating agent in the PA6 matrix. When the graphene content increased from 0 to 2.0 wt%, the crystallinity of the composites increased significantly; when it increased from 2.0 wt% to 4.0 wt%, the crystallinity did not change much. It is hypothesized that the graphene agglomerated under a high content in the matrix, which leads to an insignificant effect of the nucleating agent with a reduced surface-area-to-weight ratio. From the SEM results, it can be seen that when the graphene content is 3.0 wt% or 4.0 wt%, the lamellae appear to be stacked and agglomerated, which leads to a reduction in the specific surface area of the filler. Due to the increased crystallinity of the composite material, the two-dimensional nanosheets can hinder the motion of the polymer chains, resulting in an increase in the glass transition temperature (Tg) as the amount of graphene added increases.

### 3.2. Mechanical Properties of PA6/Gr. Nanocomposites

The impact, tensile and flexural properties of PA6/Gr. nanocomposites were tested. The results are shown in Figure 3.

As can be seen from Figure 3a,b, the impact strength of PA6/Gr. composites at both room and low temperatures was significantly improved by the addition of graphene. At the room temperature impact test, the impact strength showed a trend of increasing and then decreasing, and the maximum impact strength was 9.71 KJ/m^2^ at a 2.0 wt% graphene content, which was 130.6% higher than that of PA6. At −40 °C, the impact strength increased with the graphene addition. The maximum −40 °C impact strength was 5.83 KJ/m^2^ at a 4.0 wt% graphene content, and the most significant −40 °C impact strength increase was 5.44 KJ/m^2^ at a 2.0 wt% graphene content, which was 111.7% higher than that of PA6. The introduction of graphene improves the lamella thickness and lamellar long period, which enhances the impact resistance of the material. At 3.0 wt% and 4.0 wt% graphene contents, the SEM results shown in Figure 4 demonstrate that the graphene agglomeration and graphene stacking, as well as the uneven dispersion of graphene, may lead to stress concentration, resulting in a decrease in the impact strength performance of the PA6/Gr. composites.

The tensile strengths of PA6/Gr. composites are lower than those of PA6, as seen in Figure 3c. The tensile strength increases as the content of graphene increases. The tensile strength is lower than that of PA6 at 1.0 wt% and 2.0 wt% graphene contents, and the tensile strength is close to that of PA6 at 3.0 wt% and 4.0 wt% graphene contents. The graphene is distributed in the matrix with a different morphology, and there is a stress concentration area during the tensile process, which results in the tensile strength being smaller than that of PA6. The elongation at the break of the PA6/Gr. composites peaks at 97.8% at a 1.0 wt% graphene content, which is 131.6% higher than that of PA6. From the XRD results, it can be seen that the grain size at the graphene diffraction peak is the largest at a 1.0 wt% content, and the lamellae and matrix are tightly bonded. As the content of graphene increases, the elongation at the break shows a decreasing trend, which is inferred to be caused by the uneven distribution of graphene.

From Figure 3d, it can be seen that the flexural strength and flexural modulus of PA6/Gr. composites at 1.0 wt%, 2.0 wt% graphene contents are lower than those of PA6. At 3.0 wt% and 4.0 wt% graphene contents, the flexural strength and flexural modulus are higher than those of PA6. At a 3.0 wt% content, the flexural strength of PA6/Gr. composites is 62.04 MPa, which is 15.4% higher than that of PA6, and the flexural modulus is 2.13 GPa, which is 23.1% higher than that of PA6.

### 3.3. Microscopic Morphology of PA6/Gr. Nanocomposites

To reveal the mechanisms of the enhanced gas barrier and toughening mechanisms, the surface morphology of the material was investigated by SEM and TEM. Fracture morphology can provide detailed insights regarding filler dispersion, failure mode and filler/matrix interface [21].

The SEM images of the impact fracture surface of PA6/Gr. nanocomposites at −40 °C are shown in Figure 4. The fracture surface morphology of PA6 is relatively smooth. Graphene two-dimensional nanosheets are intercalated into the PA6 matrix. The XRD test results indicate that the grain sizes at the diffraction peaks of graphene are all larger than the van der Waals radius of helium, indicating good interfacial bonding between graphene and PA6. The impact fracture surface of PA6/Gr. composites exhibits an interlocking structure, and the two-dimensional nanosheets can hinder crack propagation. Under a larger field of view, the impact fracture surface of PA6/Gr. composites is rougher, which is an important reason for the increased toughness. When the graphene content is 1.0 wt% and 2.0 wt%, the graphene nanosheets are uniformly dispersed in the PA6 matrix under the action of shear force during extrusion processing, and the size and shape of the two-dimensional nanosheets are stable. In the direction perpendicular to the gas pressure drop, the gas diffusion path is increased, thereby enhancing the gas barrier property of the composite material. When the graphene content is 3.0 wt% and 4.0 wt%, obvious graphene sheet stacking can be observed in the low-temperature impact fracture surface, and the two-dimensional nanosheets aggregate. The dispersion of high-content graphene in the screw extrusion processing is poor, and the graphene two-dimensional nanosheets cannot be uniformly intercalated into the matrix. Graphene stacking affects the dispersion and transfer of stress in the material, leading to a decrease in impact resistance. Under identical processing conditions, high concentrations (3.0 wt%, 4.0 wt%) of graphene tend to form a stacking structure, which reduces the specific surface area of the two-dimensional nanosheets perpendicular to the gas flow path. That is, when the graphene content is 1.0 wt% and 2.0 wt%, the helium gas permeation coefficient of the composite material decreases significantly, and when the graphene content is 3.0 wt% and 4.0 wt%, the helium gas permeation coefficient changes slightly with the addition of graphene. The stacking structure in the PA6/Gr. composite leads to a decrease in the barrier effect of the two-dimensional nanosheets.

Figure 5 shows the TEM images of PA6/Gr. nanocomposites, where the two-dimensional nanosheet structure of graphene is retained in the PA6 matrix. When the graphene content is 1.0 wt% and 2.0 wt%, graphene is uniformly dispersed, and the size and shape of the two-dimensional nanosheets in the matrix are stable. When the graphene content is 3.0 wt% and 4.0 wt%, graphene stacking occurs in the matrix, leading to an inconsistent size and thickness of the sheet structures. Gaps can be observed within the stacking, which is consistent with the results of the SEM image. High-content graphene tends to aggregate and disperse unevenly in the PA6 matrix through screw extrusion processing, resulting in decreased toughness and helium barrier performance of the material.

### 3.4. Microstructure of PA6/Gr. Nanocomposites

The XRD examination of PA6/Gr. nanocomposites is shown in Figure 6. It can be seen that the crystal of PA6 is mainly of the α crystalline type, and the diffraction peaks near 20° and 23° correspond to the (200) and (002, 220) crystal planes of the crystal, respectively. Compared with PA6, the characteristic peaks of PA6/Gr. nanocomposites are stronger in intensity, indicating that the crystallinity is improved, which is consistent with the DSC results. And a new diffraction peak appears near 26.4° for PA6/Gr. nanocomposites, which corresponds to the (001) crystal plane of graphene. The intensity of the characteristic peaks increases with the increase in graphene content. The crystallite size (D) after graphene introduction was calculated by the Scheller formula [28].(2)$$D=\frac{k\lambda }{\beta \cos\theta }$$
where k is the Scheller constant (0.89); λ is the X-ray wavelength (0.1542 nm); β is the peak half width; and θ is the diffraction angle. The calculated results are listed in Table 4. It can be seen that the grain sizes at the diffraction peaks of graphene are all larger than the van der Waals diameter of helium atoms (0.28 nm), which can play a blocking role for helium atoms.

SAXS was performed on PA6 and PA6/Gr. Nanocomposites, as shown in Figure 7. The addition of graphene resulted in a shift in the peaks towards a lower q, indicating an increase in lamella spacing. The data were fitted by a one-dimensional electron density correlation function [29].(3)$$K(z)=\frac{\int_{0}^{\infty }I(q)q^{2}\cos(qz)dq}{\int_{0}^{\infty }I(q)q^{2}dq}$$

The lamellar long period (d_ac_) and the lamella thickness (d_c_) of the samples can be calculated according to Equation (2) and specific data are listed in Table 4. It can be seen that the addition of graphene increases the lamellar long period and the lamella thickness of the composites, which coincides with the enhancement in T_m_ and T_c_ in DSC. That is, the lamella thickness is elevated and the T_m_ and T_c_ temperatures are increased. The addition of graphene changes the crystalline structure of the material, and the enhancement in crystallinity, lamellar long period, and lamella thickness crystals facilitates the improvement in the gas barrier of the composites.

### 3.5. Gas Permeability of PA6/Gr. Nanocomposites

The helium barrier properties of the PA6/Gr. nanocomposites were tested, as shown in Figure 8. The gas permeability coefficient decreased as the content of graphene increased in the two-dimensional lamellar filler. From the TEM results, it can be seen that graphene was inserted into the PA6 matrix as a lamella, and the graphene nanosheets within the matrix increased the diffusion range of the gas molecular motion and had a barrier effect on the helium. The introduction of graphene acts as a nucleating agent. The crystallinity increases, the thickness of the lamellae increases, and the crystallization region also increases the diffusion range of the gas molecular motion. Together, the two-dimensional lamellar structure of the filler and the crystalline structure of the composites improved the gas barrier properties of PA6/Gr. nanocomposites.

The most significant decrease in the helium permeability coefficient was observed for a graphene content from 1.0 wt% to 2.0 wt%. The helium permeability coefficient was around 2.6 · 10^−14^ cm^3^·cm/(cm^2^·s·Pa) for a graphene content from 2.0 wt% to 4.0 wt%. The results of DSC, XRD and SAXS indicate that when the graphene content exceeds 2.0 wt%, the crystallinity of the composites and the grain size show an unobvious change. From the results of SEM, it can be seen that when the graphene content exceeds 2.0 wt%, agglomeration and stacking of graphene occur, preventing it from offering an additional projected area perpendicular to the pressure drop direction, resulting in unobvious lamellar blockage. Combining the test results of SEM, TEM, DSC and SAXS, the reasons for the improvement in the helium barrier performance of the PA6/Gr. composites can be summarized as follows: (1) graphene is uniformly dispersed in the matrix, and its two-dimensional lamellar structure is perpendicular to the direction of the gas pressure drop, which increases the diffusion range of the gas molecules; (2) graphene affects the crystalline structure of the material; with the increase in crystallinity, the lamellar long period and lamella thickness, and the increase in the crystalline size at the diffraction peaks of graphene is larger than the van der Waals diameter of helium, improving the gas barrier performance.

## 4. Conclusions

In this work, the physicochemical properties of PA6 and PA6/Gr. nanocomposites were investigated. We found that the graphene content could influence the crystalline, mechanical and gas barrier properties of PA6/Gr. nanocomposites. The following conclusions were obtained:

(1) The optimal addition of graphene was found to be 2.0 wt%, and the two-dimensional lamellar structure was oriented perpendicular to the gas pressure drop direction. At 1.0 wt% and 2.0 wt%, the graphene was dispersed uniformly in PA6, and at 3.0 wt% and 4.0 wt%, the dispersion was uneven and multilayered. The occurrence of graphene agglomeration and stacking was observed, and the specific surface area of the lamellae decreased in a direction perpendicular to the gas pressure drop.

(2) Compared with PA6, PA6/Gr. nanocomposites can crystallize at higher temperatures, with enhanced crystallization, and increased crystallinity; the lamellar long period and lamella thickness increase with the addition of more graphene content. The changes in the crystalline structure can improve the composites’ gas barrier.

(3) Compared with PA6, the impact strength of PA6/Gr. nanocomposites containing 2.0 wt% graphene at room temperature and low temperature increased from 4.21 KJ/m^2^ and 2.57 KJ/m^2^ to 9.71 KJ/m^2^ and 5.44 KJ/m^2^, respectively, and the graphene was uniformly dispersed into the matrix material as a lamellar structure, and a ligament-like phenomenon was observed. These factors likely contributed to the enhanced toughness of the composites.

(4) The helium permeability coefficient of the PA6/Gr. nanocomposite containing 2.0 wt% graphene was 2.78 · 10^−14^ cm^3^·cm/(cm^2^·s·Pa), showing a reduction of 33.2% compared to PA6, and the decrease in gas permeability was mainly attributed to the increased number of graphene flake layers perpendicular to the direction of the pressure drop, as well as to the enhanced crystallinity and increased lamellar thickness and long period of the composite.

## Figures and Tables

**Figure 1 polymers-17-00523-f001:**
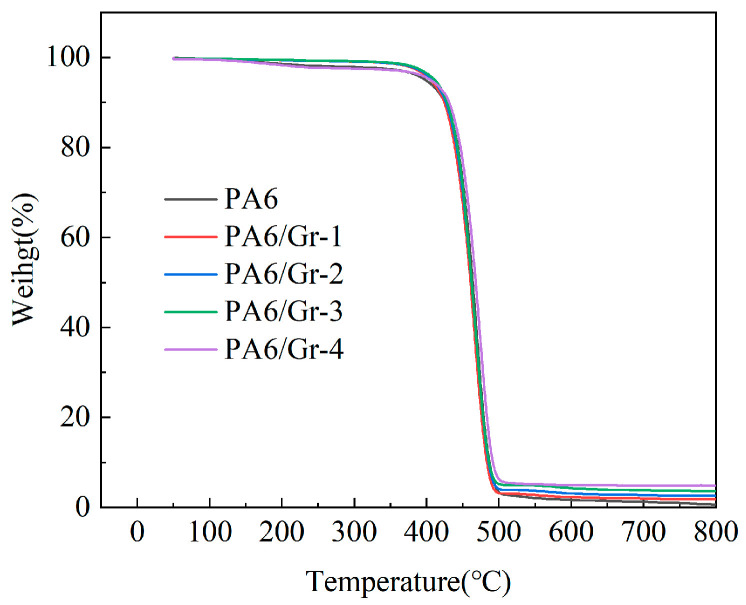
TGA curves of PA6/Gr. nanocomposites.

**Figure 2 polymers-17-00523-f002:**
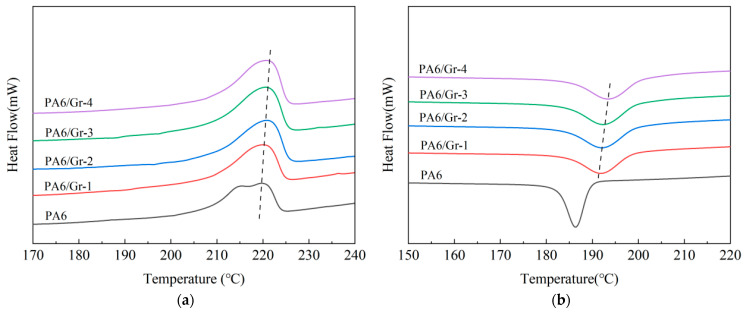
DSC curves of PA6/Gr. nanocomposites: (**a**) melting curves; (**b**) crystallization curves. (The dashed lines indicate the connections and trends of melting and crystallization peaks for different samples).

**Figure 3 polymers-17-00523-f003:**
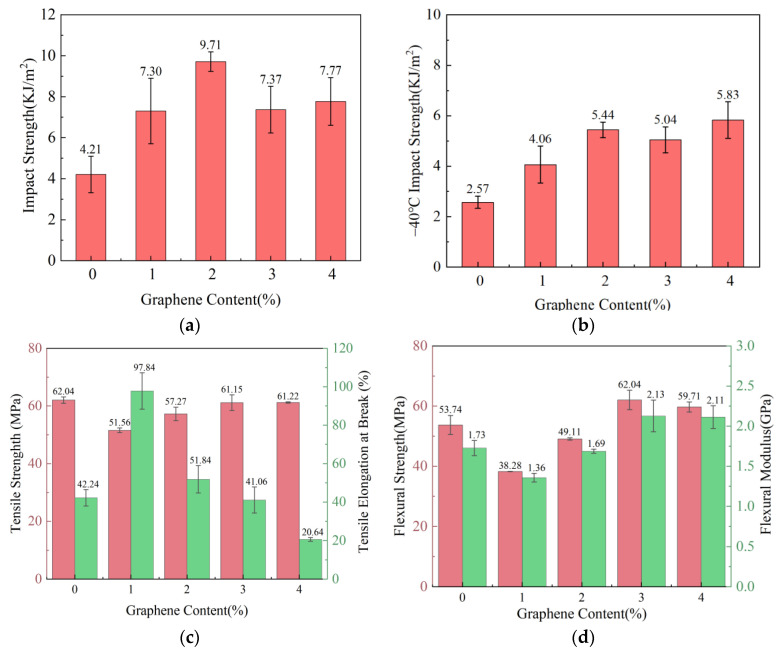
Relationship between mechanical properties and graphene content of PA6/Gr. nanocomposites: (**a**) room-temperature impact properties; (**b**) −40 °C impact properties; (**c**) tensile properties; (**d**) flexural properties.

**Figure 4 polymers-17-00523-f004:**
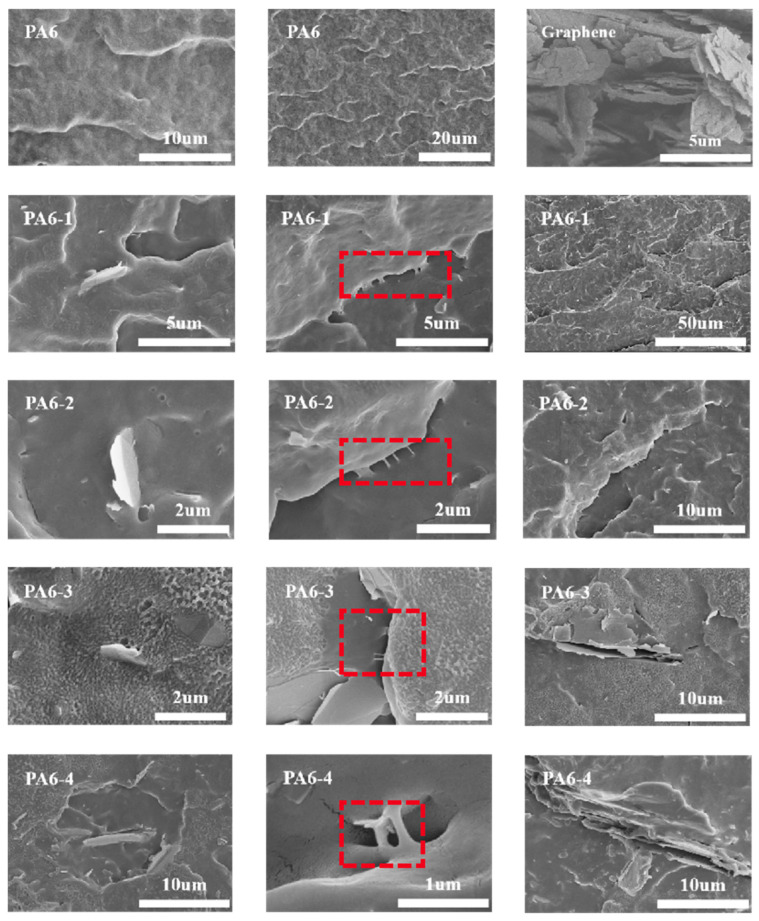
SEM image of impact section of PA6/Gr. nanocomposites. (The red dashed box indicates interlocking structure).

**Figure 5 polymers-17-00523-f005:**
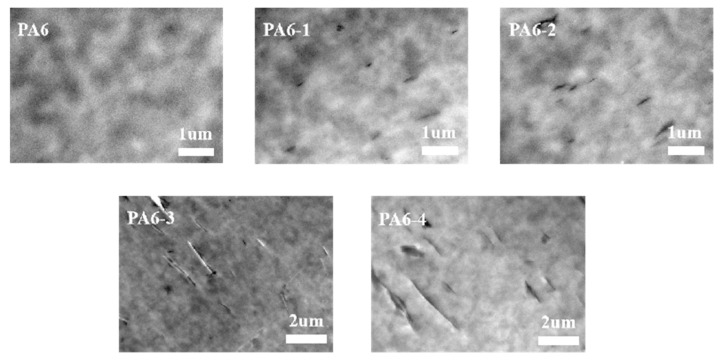
TEM image of PA6/Gr. nanocomposites.

**Figure 6 polymers-17-00523-f006:**
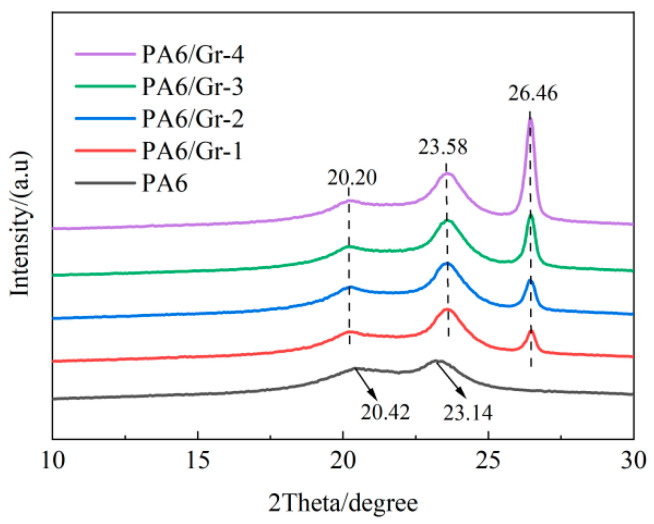
XRD pattern of PA6/Gr. nanocomposites.

**Figure 7 polymers-17-00523-f007:**
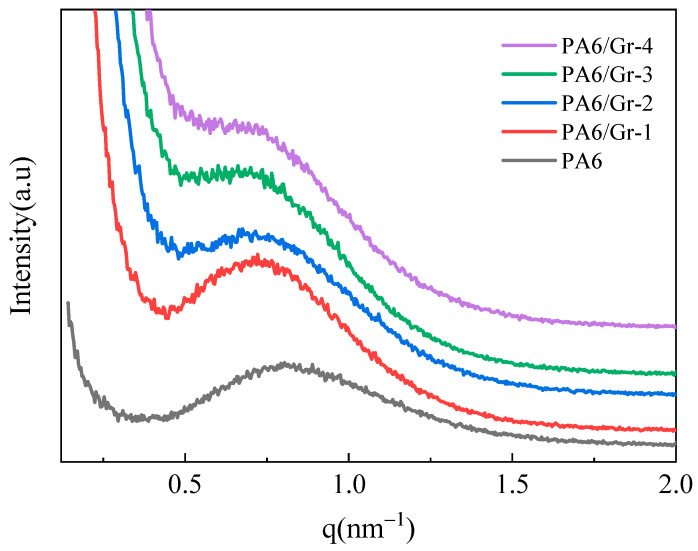
SAXS pattern of PA6/Gr. nanocomposites.

**Figure 8 polymers-17-00523-f008:**
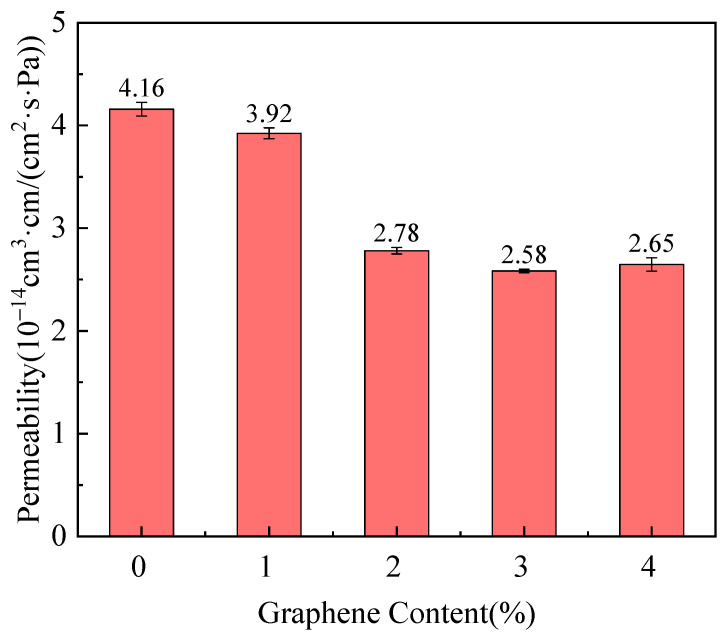
Helium permeability coefficient of PA6/Gr. nanocomposites as a function of graphene content.

**Table 1 polymers-17-00523-t001:** Composition of PA6/Gr. nanocomposites blends.

Sample	PA6/wt%	Gr./wt%	Antioxidants/wt%
PA6	99.7	0	0.3
PA6/Gr.-1	98.7	1	0.3
PA6/Gr.-2	97.7	2	0.3
PA6/Gr.-3	96.7	3	0.3
PA6/Gr.-4	95.7	4	0.3

**Table 2 polymers-17-00523-t002:** Thermal properties of materials.

Sample	Residual Mass (%)	T_5_ (°C)	T_MAX_ (°C)
PA6	0.66	395.37	455.08
PA6/Gr.-1	1.60	405.71	463.87
PA6/Gr.-2	2.60	408.91	466.38
PA6/Gr.-3	3.67	411.25	468.74
PA6/Gr.-4	4.86	413.92	471.96

**Table 3 polymers-17-00523-t003:** Crystallization and melting properties of materials.

Sample	T_c_/°C	T_m_/°C	∆H_c_ (J/g)	∆H_m_ (J/g)	Xc/%	T_g_ (°C)
PA6	186.35	219.64	55.48	42.58	18.52	54.14
PA6/Gr.-1	191.85	219.90	51.74	46.44	20.40	55.35
PA6/Gr.-2	192.15	220.80	52.82	48.42	21.48	55.83
PA6/Gr.-3	192.44	220.51	55.35	48.18	21.60	56.31
PA6/Gr.-4	193.50	220.72	53.38	48.52	21.97	56.64

**Table 4 polymers-17-00523-t004:** The samples’ grain size, lamellar long period, lamella thickness and crystallinity of the PA6 matrix and PA6/Gr. nanocomposite.

Sample	D (nm)	d_ac_ (nm)	d_c_ (nm)	$X_{C}$ (%)
PA6		7.67	3.01	18.52
PA6/Gr.-1	0.44	8.11	3.21	20.40
PA6/Gr.-2	0.38	8.11	3.21	21.48
PA6/Gr.-3	0.41	8.33	3.55	21.60
PA6/Gr.-4	0.38	8.33	3.46	21.97

## Data Availability

The original contributions presented in this study are included in this article. Further inquiries can be directed to the corresponding author.

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
