# Peer review of "Studies on Modification of Polyamide 6 Plastics for Hydrogen Storage"

_polymers, 2025, doi:10.3390/polym17040523_

Round 1

Reviewer 1 Report

Comments and Suggestions for Authors

The authors submitted their work regarding research on the modification of polyamide 6 polymers for hydrogen storage. I have examined the submitted manuscript. I have some observations and recommendations for enhancement based on its content. The article demonstrates notable advancements and is appropriate for publishing pending the resolution of the following issues, necessitating re-evaluation.

1.     The work is generally well-composed; nevertheless, certain sections, such as the introduction, may benefit from enhanced structure and clarity. Prioritize addressing the research deficit rather than reiterating talks on hydrogen energy.

2.     The potential of graphene as a nanofiller is exciting, although its novelty might be more effectively highlighted by comparisons with analogous achievements in the literature. Should alternative materials or methodologies be examined in greater depth?

3.     The rationale for the ideal graphene content (2 wt%) necessitates a more thorough examination, particularly regarding the decline in performance beyond this threshold.

4.     Could you specify the distinct advantages of your method compared to alternative two-dimensional fillers such as montmorillonite or various graphene derivatives?

5.     The influence of processing variables, such as extrusion and injection molding parameters, on the final properties of the composites has not been thoroughly investigated and requires enhancement.

6.     The practical ramifications of enhanced features, such as endurance under real-world settings, are insufficiently addressed and warrant more discussion.

7.     Have you assessed the consistency of graphene dispersion in large-scale production, and what impact does this have on the reproducibility of the results?

8.     Could you elaborate on the economic viability of utilizing graphene as a filler, specifically regarding cost-performance trade-offs? 

Comments on the Quality of English Language

There are some typographical and formatting discrepancies, particularly in references and figure captions. Please double-check the whole manuscript to avoid such issues.

Reviewer 2 Report

Comments and Suggestions for Authors

In this manuscript, the authors report on the effect of graphene addition on the mechanical and gas permeability of PA6. Up to 2 wt% of graphene, they were simultaneously improved. These results would be beneficial for the application field of hydrogen storage materials. Therefore, the reviewer recommends the publication in Polymers after addressing the following points.

1) What are type III and type IV hydrogen storage materials in the introduction?

2) The properties of PA6 and graphene were not reported in Experimental.

3) Why did T5 and Tmax decrease at high graphene loading?

4) What about the change in Tg after adding graphene?

5) Eq. (1) needs to be multiplied by 100.

6) Fig. 3(d) was erroneously replaced by Fig. 3(c).

Round 2

Reviewer 1 Report

Comments and Suggestions for Authors

The revision meets the reviewer's requirements, so the revised manuscript can be accepted for publication.